# Real-time, spatial decision support to optimize malaria vector control: The case of indoor residual spraying on Bioko Island, Equatorial Guinea

**Guillermo A. García**[1], **Brent Atkinson**[1], **Olivier Tresor Donfack**[2], **Emily R. Hilton**[3], **Jordan M. Smith**[2], **Jeremías Nzamío Mba Eyono**[2], **Marcos Mbulito Iyanga**[2], **Liberato Motobe Vaz**[2], **Restituto Mba Nguema Avue**[2], **John Pollock**[4], **Josea Ratsirarson**[1], **Edward M. Aldrich**[4], **Wonder P. Phiri**[2], **David L. Smith**[3], **Christopher Schwabe**[4], **Carlos A. Guerra**[1] *

1 Medical Care Development International, Silver Spring, MD, United States of America, 2 Medical Care Development International, Malabo, Equatorial Guinea, 3 Institute for Health Metrics and Evaluation, Univeristy of Washington, Seattle, WA, United States of America, 4 Medical Care Development, Augusta, ME, United States of America

* cguerra@mcd.org

**Data Availability Statement:** Full IRS coverage data used in the analyses (2017-2021) are available at https://doi.org/10.6084/m9.figshare.16688803.

## Abstract

Public health interventions require evidence-based decision-making to maximize impact. Spatial decision support systems (SDSS) are designed to collect, store, process and analyze data to generate knowledge and inform decisions. This paper discusses how the use of a SDSS, the Campaign Information Management System (CIMS), to support malaria control operations on Bioko Island has impacted key process indicators of indoor residual spraying (IRS): coverage, operational efficiency and productivity. We used data from the last five annual IRS rounds (2017 to 2021) to estimate these indicators. IRS coverage was calculated as the percentage of houses sprayed per unit area, represented by 100x100 m *map-sectors*. *Optimal coverage* was defined as between 80% and 85%, and *under* and *over-spraying* as coverage below 80% and above 85%, respectively. *Operational efficiency* was defined as the fraction of map-sectors that achieved optimal coverage. Daily productivity was expressed as the number of houses sprayed per sprayer per day (h/s/d). These indicators were compared across the five rounds. Overall IRS coverage (*i.e.* percent of total houses sprayed against the overall denominator by round) was highest in 2017 (80.2%), yet this round showed the largest proportion of oversprayed map-sectors (36.0%). Conversely, despite producing a lower overall coverage (77.5%), the 2021 round showed the highest operational efficiency (37.7%) and the lowest proportion of oversprayed map-sectors (18.7%). In 2021, higher operational efficiency was also accompanied by marginally higher productivity. Productivity ranged from 3.3 h/s/d in 2020 to 3.9 h/s/d in 2021 (median 3.6 h/s/d). Our findings showed that the novel approach to data collection and processing proposed by the CIMS has significantly improved the operational efficiency of IRS on Bioko. High spatial granularity during planning and deployment together with closer follow-up of field teams

v1. These data were aggregated at the map-sector-level to avoid privacy issues.

**Funding:** The author(s) received no specific funding for this work.

**Competing interests:** The authors have declared that no competing interests exist.

using real-time data supported more homogeneous delivery of optimal coverage while sustaining high productivity.

## Author summary

Effective public health interventions rely on high coverage to provide community protection. Coverage is determined by the proportion of a given target population that receives the intervention. The level of coverage required varies across settings and health problems. The question about how one achieves high coverage in an equitable manner is operationally challenging. Here, we describe the use of digital tools to support and optimize the delivery of a crucial and proven malaria control intervention, indoor residual spraying (IRS), on Bioko Island. We demonstrate that the scale at which one plans delivery and calculates coverage is critical for guaranteeing that the whole target population is served equally. We also show that achieving adequate high coverage during IRS implementation is challenging, but can be greatly supported by subdividing the target area into multiple, small area units and by using spatial decision support to guide deployment. We focused on IRS as a specific example, but the same digital tools can be used for other public health interventions, with an approach that promotes decision-making during implementation and allows better monitoring of intervention coverage, resulting in more efficient delivery.

## Introduction

Public health precision requires efficient and effective targeting of interventions to those most in need using the best available evidence [1–3]. Spatial decision support systems (SDSS) represent critical tools to achieve this goal by transforming disease data into information and knowledge useful for decision-making [3]. The spatial component is essential to enable the prioritization of resources and efficient and equitable delivery of interventions. This is particularly relevant for malaria vector control interventions that aim to provide community-wide protection.

Indoor residual spraying (IRS) is a critical component of malaria control in many endemic countries [4, 5]. IRS delivery is a challenging endeavour that entails the simultaneous deployment of many fieldworkers within a geographical area. Often, malaria programmes plan and deploy IRS based on a target daily productivity per sprayer [6, 7]. This demands close and strategic management and monitoring of activities. The ultimate goal of IRS is to achieve universal coverage that assures community protection [8–10].

Canonically, the recommended threshold for universal IRS coverage has been loosely defined as between 80% and 85% of houses sprayed within a given targeted area [7, 11]. Though the evidence supporting this recommendation is limited [12–14], IRS is an expensive intervention [15–17], and there is a need to weigh the community effects of IRS against maximizing coverage equity. Striking this balance could maximize the overall impact of scarce resources. Operationally, this would require optimizing the use of commodities and labor by maximizing productivity towards reaching optimal coverage based on known denominators. But denominators are often unknown and at times inaccurate.

The latter was an early challenge when IRS was instated as the principal vector control strategy on Bioko Island in 2004. The ever-changing household denominator, particularly in the

rapidly growing urban areas of the island [31], posed significant problems for estimating and achieving coverage targets. Consequently, coverage and fieldworker productivity estimates were misleading. This prompted investing significant efforts not only in establishing accurate denominators but also in precisely measuring these critical IRS process indicators. S1 Text provides deeper context into IRS operations on Bioko Island.

We refer to *optimal operational coverage* (hereafter *optimal coverage*) as a key concept for measuring and interpreting IRS coverage and articulate it with a simple thought experiment (see Fig 1 and Box 1). Optimization in this context refers to the need of guaranteeing homogeneous, high coverage across all targeted areas using limited resources. Heterogeneity in coverage can potentially leave gaps of unprotected populations despite seemingly achieving overall adequate coverage, a phenomenon explained by the scale effect and subject to concerns surrounding the modifiable areal unit problem (MAUP) [18, 19]. We also introduce the term *operational efficiency* to express the frequency at which optimal coverage is achieved, which would also improve the equitable delivery of IRS, maximizing its community effect. This is a non-trivial undertaking that could be operationalized through the use of a SDSS.

SDSS have long been recognized as a necessity for malaria control and elimination. During the Global Malaria Eradication Programme, geographical reconnaissance was advocated as essential for the attack and consolidation phases to assure that interventions reached every household [20]. At the time, digital tools were both incipient and not readily available to complement paper maps with data. Early efforts to improve intervention management using information technology were documented for southern Africa [21]. This initiative used relational databases and geographic information systems to replace paper-based reports, facilitating regular monitoring of spray coverage, worker performance and insecticide use. Data systems enabled the resolution of common operational problems during implementation. The further advancement of information technology allowed SDSS to gain considerable traction in current malaria control and elimination programmes, particularly regarding the delivery of vector interventions [22–28]. Real-time data support is also increasingly acknowledged as essential for malaria monitoring and surveillance [29, 30].

The aim of this paper is to describe how the use of a novel SDSS to support malaria control on Bioko Island has affected IRS coverage and operational outcomes overtime. Given its operational scope, the paper does not include a description of the impact of IRS operations on malaria prevalence. Malaria epidemiology is affected by multiple exogenous factors that add significant complexity and these are subject of ongoing investigations on Bioko. Such analyses fall beyond the reach of the current paper. The SDSS in question, the Campaign Information Management System (CIMS), is fully described in S2 Text and S1 Fig. The CIMS has been progressively developed and used to support malaria control on Bioko. The grid-based mapping system underpinning the CIMS is described in detail elsewhere [31] as well as in S2 Text and S2 Fig, and represents the crux behind intervention deployment planning. We used data from the last five annual IRS rounds on Bioko Island (2017 to 2021) generated through the CIMS to track coverage and productivity. Over this period, the CIMS evolved from providing accurate denominators for planning at the community level (2017 and 2018), to deploying interventions at the more granular, 100x100m map-sector-level [31] and following teams in real-time (2019 to 2021). This five-year period, thus, offers a snapshot of the learning-by-doing process of utilizing the CIMS. For context purposes, S1 Text includes a brief account of targeting and deploying IRS on Bioko Island.

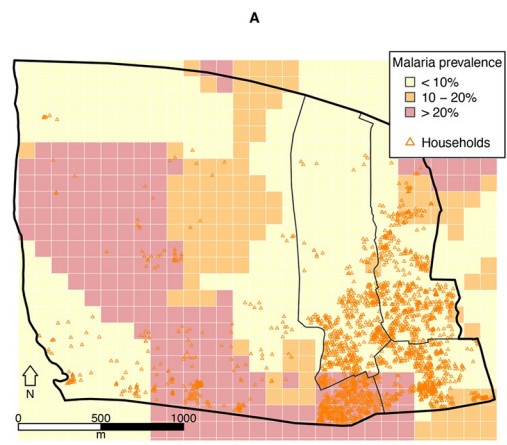

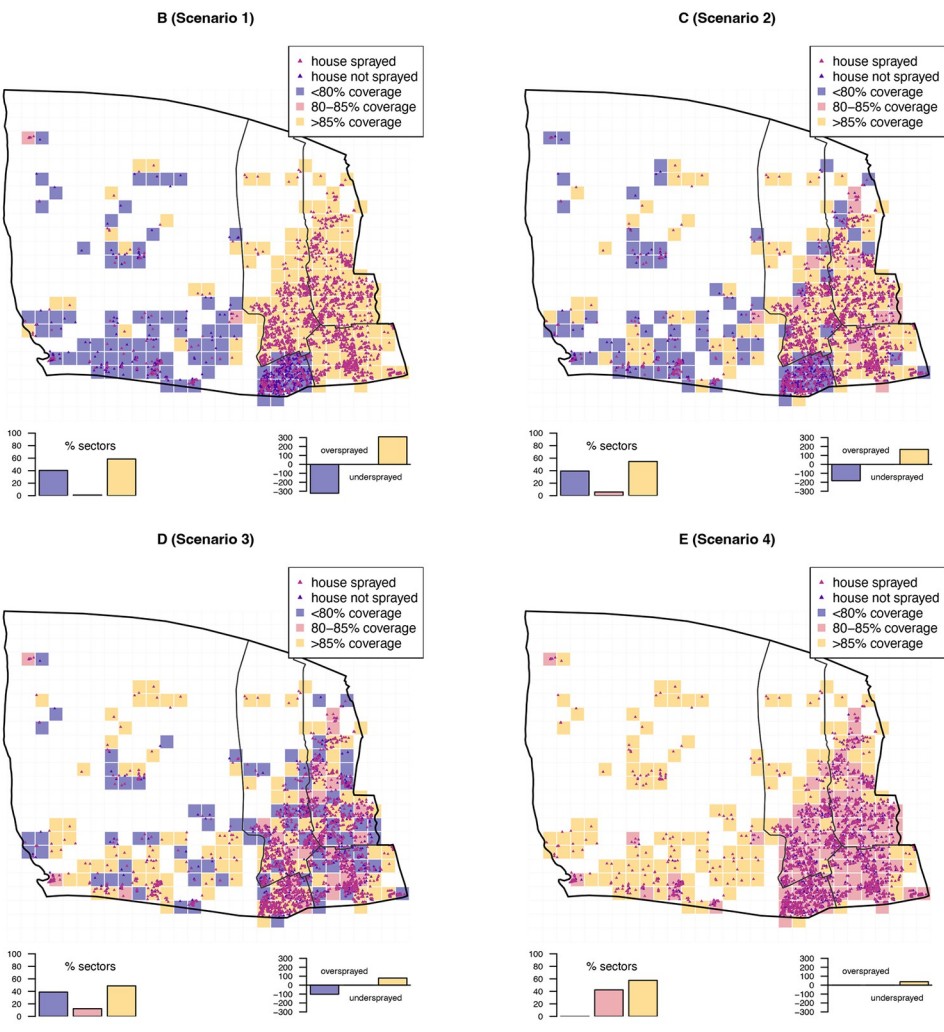

**Fig 1. Achieving optimal spray coverage through planning at higher spatial granularity. A**. An administrative division of Bioko Island, its four subdivisions and the distribution of houses within (n = 2,341); overlaid are the map-sectors (n = 203) and a smoothed malaria prevalence surface. **B-E**. Different hypothetical scenarios of IRS, all achieving optimal coverage at the administrative division-level. In scenarios 1 to 3, 1,873 houses are sprayed to achieve exactly 80% coverage in different configurations. In scenario 1, all the houses in three subdivisions and 16.8% of houses in the

fourth subdivision are sprayed. In scenario 2, 90% of houses in three subdivisions and 48.5% in the fourth are sprayed. In scenario 3, 80% of randomly selected houses across the division are sprayed. In scenario 4, spraying is deployed based on optimal coverage calculated at map-sector-level, with 1,946 houses sprayed and an overall 83.1% coverage. The bar graphs depict the proportion of map-sectors by coverage (left) and the number of houses over and undersprayed (right) in each scenario. The latter refer to the number of houses that were unnecessarily sprayed and those which should have been sprayed in order to reach optimal coverage, respectively.

## Materials and methods

### Data

The data for this study were gathered using the CIMS (S2 Text and S1 Fig). Briefly, the CIMS is an Android-based, open-source application that leverages PostgreSQL (The PostgreSQL Global Development Group) for data storage, management and analytics. Fieldworkers capture data offline on tablets using dedicated forms. Once linked to the Internet, the data are transferred to the server. All field activity parameters, campaign settings and data for calculating output indicators are set up on the server, from which they are downloaded and analyzed using visualization software (Tableau Software, LLC, WA, USA) to assist in decision-making. The CIMS is used at all stages of IRS operations, from keeping the household denominator up-to-date (*i.e.* updating the household status during visits of any intervention) to planning (*i.e.* which map-sectors to spray based on malaria prevalence, allocating the work of field teams equitably along a deployment plan, etc.) to monitoring progress and guiding teams (*i.e.* through the use of real-time data to track coverage and productivity; S3 Fig) to evaluating the operation (*i.e.* process and impact indicators).

We used data from the last five annual rounds of IRS on Bioko (2017 to 2021). Though the intervention has been implemented since malaria control operations began on the island in 2004, the CIMS was gradually introduced since 2014 but used more rigorously only since 2017. Since 2015, the IRS strategy shifted from island-wide deployment to targeting the highest burden communities. This change was motivated by budget constraints and by the malaria control program also relying on mass bed net distributions and focal larviciding. Before 2021, the aim throughout all rounds was to spray 80% of houses. In response to increasing malaria prevalence, the 2021 IRS round once again aimed to spray the whole island, but only around half of the map-sectors were targeted at 80% coverage. This is part of an ongoing operational trial that attempts to measure the impact of different IRS coverage thresholds (unpublished work). For comparison with the four preceding rounds, only the 80% coverage map-sectors were included in the analyses.

### Calculating coverage

For each round, IRS coverage was estimated overall (*i.e.* dividing the total number of houses sprayed in the round by the total number of inhabited houses in all targeted map-sectors) and at each map-sector (*i.e.* dividing the number of houses sprayed by the number of inhabited houses within the map-sector). Operational efficiency, overspraying, and underspraying were defined based on the 80% to 85% optimal coverage band (see Box 1) and map-sectors were classified accordingly. For map-sectors with very small denominators, overspraying is inevitable. In such map-sectors, achieving optimal coverage necessarily means spraying 100% of houses, biasing the interpretation of our results. To accommodate this bias, for our map-sector coverage calculations we included only map-sectors with denominators equal or higher than a convenient cutoff of 10 houses.

Box 1

## Optimizing IRS coverage

We define a band for *optimal coverage* as no less than 80% and no more than 85% of houses sprayed in a given map-sector, pragmatically justified on the basis of established canons rather than on existing evidence [7]. It is plausible that optimal upper and lower bounds for coverage may well be set different and likely context-specific, depending on heterogeneous transmission and programmatic goals. We use a band rather than a single cutoff because often it is practically impossible to spray an exact proportion of the denominator. Any coverage below and above this band represents *under* and *overspraying*. These concepts are motivated by operational rather than epidemiological principles, as the goal is to balance community protection against use of resources. The assumption is that resources are limited and commodities are procured to maximize cost-effectiveness while assuring community-wide protection of the entire population targeted for the intervention.

Fig 1 illustrates these concepts using four hypothetical scenarios within an administrative division of Bioko Island (Fig 1A). The details of each scenario are explained in the caption of Fig 1. In this example, we assume that sufficient insecticide and human resources are secured for spraying between 80% and 85% of houses within the administrative division, which was selected to receive IRS based on predetermined criteria.

Scenario 1 is the worst-case because it leaves a large proportion of map-sectors under-sprayed (40.4%), failing to achieve the community protection objective while also over-spraying 58.6% map-sectors and thus failing the cost-effectiveness objective (Fig 1B). Scenario 2 has a lower level of over and underspraying (54.7% and 39.4% map-sectors over and undersprayed, respectively; Fig 1C). These scenarios could be expected when spray teams are guided through a deployment designed for convenience and logistical ease at the expense of coverage and resources. One explanation could be that houses in the East of the administrative division are more accessible than those in the West. In scenario 3, under and oversprayed map-sectors are interspersed throughout the administrative division. Despite this, 38.8% and 48.8% of map-sectors are under and oversprayed, respectively, though fewer houses within these map-sectors were sprayed above the required number to achieve optimal coverage (Fig 1D).

In all three scenarios, a low proportion of map-sectors are optimally covered (1% in scenario 1, 5.9% in scenario 2 and 12.3% in scenario 3). By way of contrast, in scenario 4, 42.4% of map-sectors are adequately covered and no map-sectors have coverage under 80%. Even though 57.6% of map-sectors in scenario 4 are oversprayed, this is explained by the small number of houses within them. This translated into only 38 houses sprayed over the number required to achieve optimal coverage in these map-sectors. All of this is achieved with an overall IRS coverage of 83.1%, or 73 more houses sprayed than in the other scenarios (Fig 1E).

An important consideration is that malaria transmission is highly heterogeneous [45, 49, 50]. Fig 1A illustrates this heterogeneity, where the highest malaria prevalence appears localized in 63 map-sectors, mainly along the West and the South (pink map-sectors in Fig 1A where *Pf*PR > 20%). It follows that blanket intervention deployment, such as scenarios 1, 2 and 3, misses protecting populations at the highest risk and, depending on

how IRS affects mosquito ecology, could fail to maximize the community benefits in surrounding areas. This is particularly the case in scenarios 1 and 2, in which 79.4% and 68.3% of high *Pf*PR map-sectors have sub-optimal IRS coverage. In scenario 3, although a more even coverage is achieved, 39.7% of the high prevalence map-sectors are undersprayed. In scenario 4, universal coverage is achieved, protecting the entire population at risk of malaria, including all those living within the highest malaria risk map-sectors.

These theoretical scenarios are not accounting for the spill-over effects of interventions such as IRS, whereby populations inhabiting map-sectors adjacent to those sprayed are also protected. They also assume that optimal spraying is defined by the 80–85% band, when in reality these bounds may be lower or higher, depending on the setting. Different assumptions would change the way these results are interpreted or even how the intervention is deployed in the first place. Notwithstanding this caveat, the scenarios serve as a stark reminder that intervention deployment can be severely biased by the geographic scale at which coverage is calculated. This bias is due to the scale effect of the MAUP [18, 19].

We determined the number of houses needed to spray in each map-sector in order to achieve optimal coverage. This was calculated as the ceiling of *denominator* * .8 and as the floor of *denominator* * .85, with the rule that the latter would have to be equal to or higher than the former. For example, for a map-sector with 25 inhabited houses, the number of houses to spray would be between 20 and 21 (*i.e.ceiling*(25 * .8) and *floor*(25 * .85)). We then calculated *relative coverage* at each map-sector, which corresponded to the ratio of houses sprayed to houses needed to spray to obtain optimal coverage according to the above calculation. In the example above, relative coverage was 1 if the houses sprayed in that map-sector were 20 or 21. Map-sectors were classified as optimally, over or undersprayed if their relative coverage was equal to, above or below 1, respectively.

We calculated underspraying and overspraying at two different scales. First, the number of houses sprayed below and over the optimal coverage band within each map-sector was aggregated for all targeted map-sectors to provide the overall number of *houses* that were under and oversprayed per round. Second, the number of map-sectors at, below and above optimal coverage represented the number of optimally sprayed, undersprayed and oversprayed *map-sectors*. Both the number of houses and map-sectors thus classified were expressed as the proportions of the total houses and map-sectors sprayed, respectively, and were compared between rounds using tests of proportions in R [32].

## Calculating productivity

Productivity was measured as the number of houses sprayed per spray operator per day (h/s/d). According to the World Health Organization, productivity is expected to be as high as 10 to 15 h/s/d in locations where houses are easily accessible and relatively small, and as low as 5 h/s/d where houses are scattered or large [7]. In the landscape of Bioko Island, IRS operational challenges differ by setting, which includes approximately 80% urban areas and 20% periurban and rural areas. Households in urban areas of Bioko present high refusal rates (S4 Fig), low availability rates and often reduced co-operation by residents to prepare their house for spraying. This inevitably reduces the average daily productivity attainable by sprayers. Based on experience across the many annual rounds of IRS, minimum target productivity for

**Table 1. Houses sprayed in the last five rounds of IRS on Bioko Island (2017–2021).** Den = Denominator; Cov = Coverage; Under = Undersprayed; Over = Over-sprayed; Prod = Productivity. Percentages and inter-quartile ranges (IQR) are within square brackets. [†]Percent relative to the number of houses sprayed. [‡]Productivity is expressed as the median and IQR of houses sprayed per sprayer per day. Statistical significance of differences with the preceding round were determined by $\alpha < 0.05$ in the test of proportions (marked [**]) and in the Wilcoxon-Mann-Whitney test (marked [*]).

| IRS round | Den (n) | Sprayed (n) | Cov (%) | Under (n) [%[†]] | Over (n) [%[†]] | Prod[‡] (h/s/d) [IQR] |
|---|---|---|---|---|---|---|
| 2017 | 21,900 | 17,563 | 80.2 | 1,271 [7.2] | 566 [3.2] | 3.4 [1.9–4.3] |
| 2018 | 21,184 | 16,613 | 78.4 | 1,609 [9.7][**] | 453 [2.7] | 3.8 [2.2–4.8] |
| 2019 | 19,978 | 15,793 | 79.1 | 1,208 [7.6][**] | 362 [2.3] [**] | 3.8 [2.9–4.6] |
| 2020 | 43,587 | 32,537 | 74.6 | 4,035 [12.4][**] | 529 [1.6][**] | 3.3 [2.6–3.9][*] |
| 2021 | 31,804 | 24,662 | 77.5 | 1,982 [8.0][**] | 291 [1.2][**] | 3.9 [2.8–4.5][*] |

planning purposes on Bioko has been defined at 4 h/s/d. We reported productivity using the median and inter-quartile range of the daily productivity by round and compared productivity between rounds using the Wilcoxon-Mann-Whitney non-parametric test [32].

## Results

### IRS coverage

Table 1 presents the overall summaries of houses sprayed by round. A total of 107,168 houses were sprayed in the five years, with a mean of 21,437 houses per year and almost a third of which were sprayed in 2020 (32,537). Considerably more houses were sprayed in 2021 (43,382), but 18,720 (43.2%) were excluded from the analyses given they were located within map-sectors where coverage targets were lower than 80% (see Methods). Overall coverage was highest in 2017 (80.2%) and lowest in 2020 (74.6%). The latter was largely explained by the challenges presented for IRS operations during the establishment of the COVID-19 pandemic [33].

The fraction of undersprayed and oversprayed houses was expressed as the total number of houses below and above the number needed to spray (see Methods). This showed that the fraction undersprayed was highest in 2020, with 4,035 houses (12.4% of the total houses sprayed) short of the target needed to reach 80% coverage. Underspraying was relatively low in 2017 and 2019, with 1,271 (7.2%) and 1,208 (7.6%) houses. This coincided with both these rounds achieving the highest overall coverage. In 2017, overspraying was higher than in all other rounds (566 houses, representing 3.2% of all houses sprayed in that round). Overspraying decreased progressively across the four succeeding rounds to 1.2% in 2021 (Table 1). Significant reductions in overspraying were seeing from 2019 onwards.

Considering under and overspraying by map-sectors provided a clear picture of operational efficiency and the factors driving it across rounds (Table 2; Figs 2, 3 and 4A). Spraying took

**Table 2. Map-sectors sprayed in the last five rounds of IRS on Bioko Island (2017–2021).** Percentages are within square brackets. Operational efficiency refers to counts and percentages of map-sectors where optimal coverage was achieved. [**]Statistically significant difference with the preceding round, determined by $\alpha < 0.05$ in the test of proportions.

| IRS round | Denominator (n) [%] | | Operational efficiency (n) [%] | | Undersprayed (n) [%] | | Oversprayed (n) [%] | |
|---|---|---|---|---|---|---|---|---|
| | All | ≥ 10 houses | All | ≥ 10 houses | All | ≥ 10 houses | All | ≥ 10 houses |
| 2017 | 1,414 | 628 [44.1] | 549 [38.8] | 129 [20.5] | 554 [39.2] | 273 [43.5] | 311 [22.0] | 226 [36.0] |
| 2018 | 1,751 | 552 [31.5] | 778 [44.4][**] | 125 [22.6] | 709 [40.5] | 264 [47.8] | 264 [15.1][**] | 163 [29.5][**] |
| 2019 | 1,205 | 555 [46.1] | 508 [42.2] | 126 [22.7] | 477 [39.6] | 270 [48.6] | 220 [18.3][**] | 159 [28.6] |
| 2020 | 2,226 | 1,088 [48.8] | 709 [31.9][**] | 194 [17.8][**] | 1,217 [54.7][**] | 677 [62.2][**] | 300 [13.5][**] | 217 [19.9][**] |
| 2021 | 1,835 | 818 [44.6] | 785 [42.8][**] | 308 [37.7][**] | 838 [45.7][**] | 357 [43.6][**] | 212 [11.6][**] | 153 [18.7] |

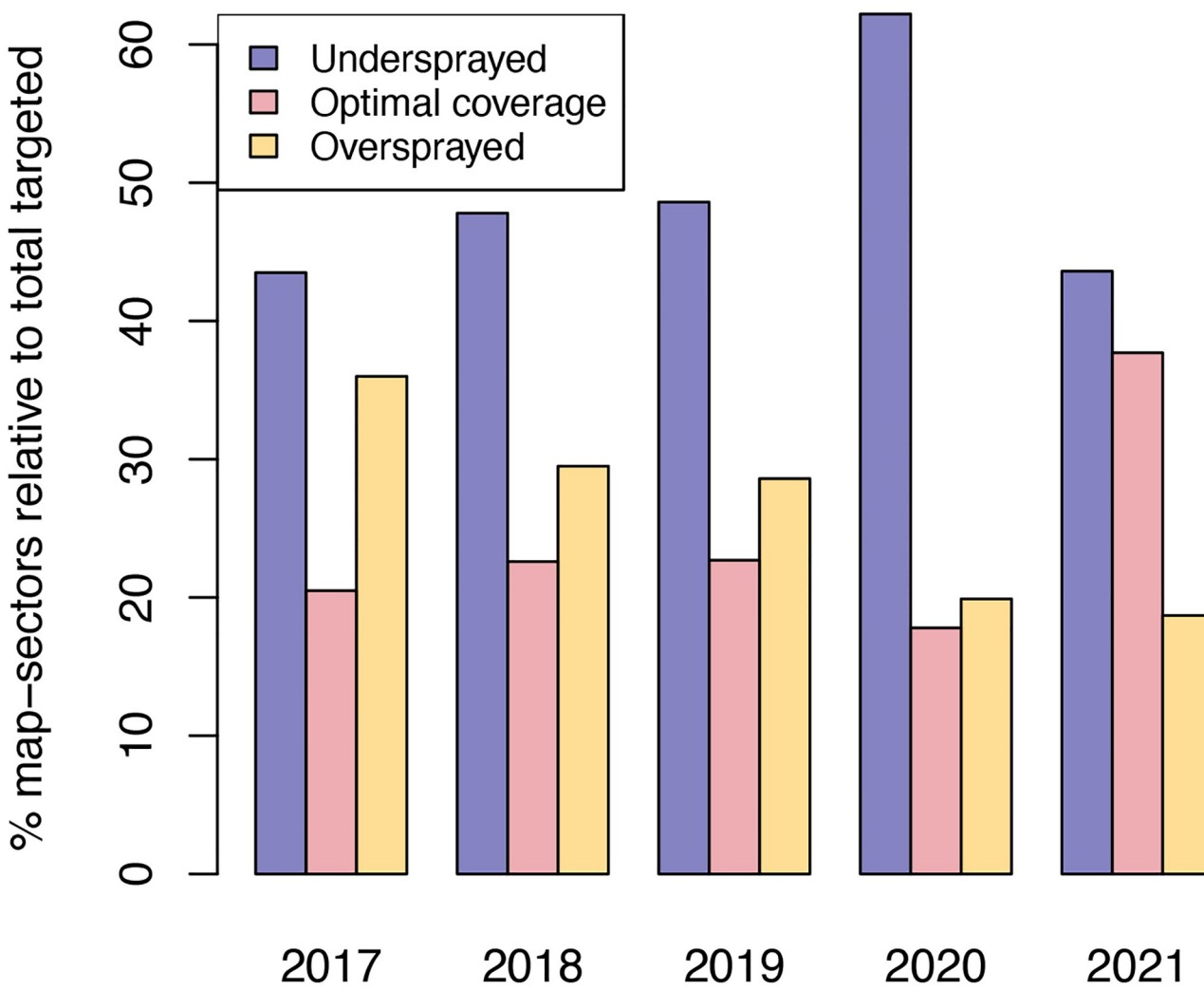

**Fig 2. Map-sector-level IRS coverage in the last five rounds on Bioko Island.** Only data for large denominator map-sectors ($\geq$ 10 houses) are depicted.

place in an average 1,686 map-sectors per round, between a third and a half of which had denominators of 10 or more houses. As was mentioned in the Methods, excluding sparsely occupied map-sectors allowed a more accurate assessment of the three different levels of coverage. Therefore, the results presented below and in Figs 2, 3 and 4A refer to large denominators only.

Fig 2 shows the proportion of map-sectors at each coverage category. The proportion of undersprayed map-sectors was highest in 2020 (62.2%), a statistically significant increase ($P < 0.001$) before dropping to 43.6% in 2021. Between 2017 and 2019, the differences in the proportions of undersprayed map-sectors were not significant, but the reduction in this percentage between 2019 (48.6%) and 2021 (43.6%) was statistically significant ($P = 0.038$). Operational efficiency was similar from 2017 to 2019, when the target coverage was achieved only in about a fifth of the sprayed map-sectors. In 2020, operational efficiency dropped significantly (17.8%, $P = 0.011$) before increasing significantly to 37.7% in 2021 ($P < 0.001$). The anomalies

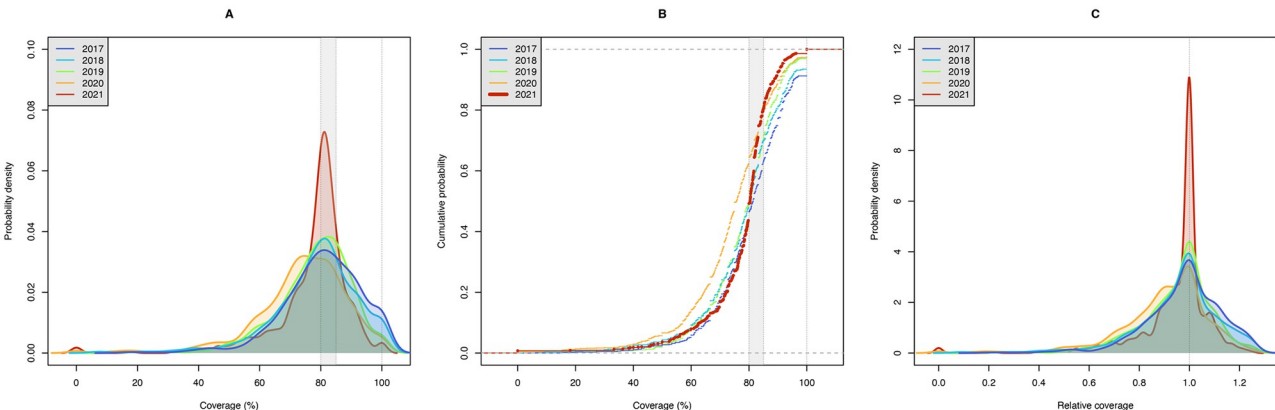

**Fig 3. Distribution of IRS coverage in large denominator map-sectors (≥ 10 houses) on Bioko Island in the last five rounds (2017 to 2021). A.**
Probability density functions of coverage, by round. **B**. Cumulative density functions of coverage, by round. The grey band marks the optimal coverage
range between 80% and 85%. A vertical line at the maximum coverage of 100% is also drawn to highlight the level of overspraying. **C**. Probability
density functions of relative coverage, by round. Relative coverage is calculated by the ratio of actual houses sprayed to houses needed to spray to
achieve no less than 80% and no more than 85% coverage, where 1 is equivalent to optimal coverage (see main text).

observed in 2020 were attributed to disruptions caused by the COVID-19 pandemic on pro-
ductivity [33].

Overspraying of map-sectors progressively decreased in the five-year period, from a high of
36.0% in 2017 to a low of 18.7% in 2021. There was a significant reduction in map-sector over-
spraying in 2018 (29.5%, $P = 0.011$) and then another in 2020 (19.9%, $P < 0.001$). Again, the
latter could be explained by the lower overall productivity and coverage in that year due to
COVID-19 [33], but this drop was sustained in 2021 with a further, though not significant
($P = 0.268$), reduction to 18.7%.

Figs 3 and 4A illustrate the distribution of IRS coverage across map-sectors in all rounds.
The probability density and the cumulative distribution functions show that the 2021 round
was considerably more operationally efficient, reducing both over and underspraying in map-
sectors. The sharp spike of relative coverage in Fig 3C is a clear indicator of this achievement,
further reinforced in Fig 4A, where the violin plot of the 2021 round appears considerably
wider around relative coverage = 1.

## IRS productivity

Median productivity across all rounds was 3.6 h/s/d, ranging from 3.3 in 2020 to 3.9 h/s/d in
2021 (Table 1). No significant differences were observed between rounds, except for 2020,
when the drop in productivity was statistically significant ($P < 0.001$) followed by a significant
increase in 2021 ($P < 0.001$). There was no statistically significant difference between produc-
tivity in 2019 and in 2021 ($P < = 0.608$). Fig 4B illustrates the distribution of daily productivity
by round. In 2017 and 2018, productivity was over-dispersed, but this distribution narrowed
progressively around the target of 4 h/s/d in the most recent rounds. Fig 4 serves as an illustra-
tion that despite improvements in productivity were only marginal this was concomitant with
significant improvements in operational efficiency.

## Discussion

Using data from IRS rounds, we examined how a customised SDSS has positively impacted
operations on Bioko Island. This study focused exclusively on process indicators as we sought

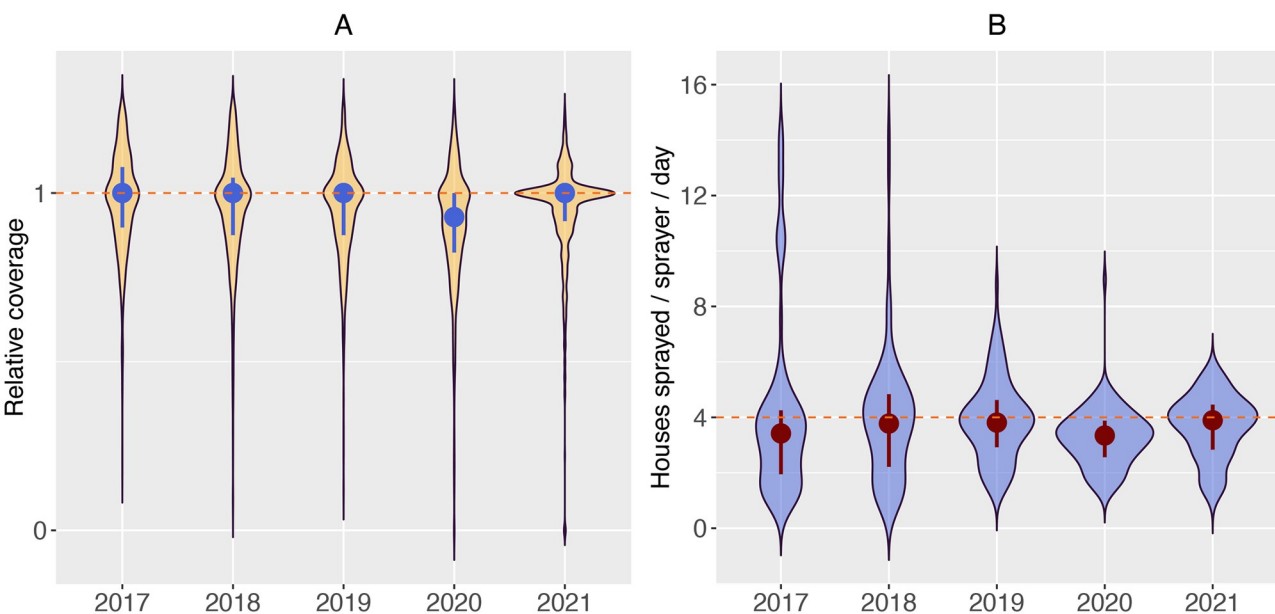

**Fig 4. Optimal coverage in large denominator map-sectors ($\geq$ 10 houses) and productivity in the last five IRS rounds on Bioko Island (2017 to 2021). A**. Distribution of relative coverage across large denominator map-sectors. **B**. Productivity by round, measured as the number of houses sprayed per sprayer per day. The distribution corresponds to the daily productivity throughout each round. The circle and vertical lines in the middle of each violin plot denote the median and inter-quartile ranges. The dashed horizontal lines mark optimal coverage and the target productivity of 4 h/s/d for IRS deployment on Bioko Island.

to investigate coverage and productivity gains. The evaluation of impact will be addressed elsewhere as part of comprehensive analyses of malaria epidemiology on Bioko Island and of how the long history of vector control and a plethora of other drivers have shaped transmission dynamics.

Data from the last five rounds of IRS showed that implementation, supported by the CIMS, has improved operational efficiency while maintaining adequate productivity (*i.e.* based on the 4 h/s/d productivity target used for operational planning on Bioko Island). Operational efficiency increased significantly, almost doubling, from 20.5% in 2017 to 37.7% in 2021. This improvement was attained by significantly reducing overspraying by about half, from 36.0% of oversprayed map-sectors in 2017 to 18.7% in 2021. Similar trends in underspraying were not observed, with proportions in 2017 resembling those in 2021. The 2020 IRS round was a notable outlier, with significantly less optimal coverage and more undersprayed map-sectors. This was ascribed mainly to hurdles presented during that year due to the impact of the COVID-19 pandemic [33]. This anomaly aside, our analyses show that achieving optimal coverage is challenging regardless of improved IRS monitoring through an SDSS.

Persistent underspraying could have been driven by two main, difficult to control barriers that are independent of deployment. First, there is refusal by the community, which is a commonly encountered impediment to IRS campaigns in many endemic countries [34]. Poor community acceptance is a recurrent problem in parts of Bioko Island, particularly in urban areas. Ongoing investigations are looking at characterizing refusal patterns, their reasons and how to tackle them with improved communication strategies (see S4 Fig). Second, accessibility to households during spray rounds can be hindered by absenteeism, which is often the case in urban communities of Malabo, where household heads are not present during IRS working hours.

Managing overspraying is more reliant on planning and implementation. As such, it is imperative that spray teams have access to data that signal when coverage targets have been reached, prompting exit from the map-sector. Evidently, this protocol has improved over the last years as is reflected by the significant decrease in overspraying. It is not uncommon, however, to find situations when residents request fieldworkers to spray their houses even when the target has been met. Faced with such circumstances, field teams always prioritize requests of residents over coverage goals. This could partly explain why, despite the real-time support of the CIMS, some map-sectors were oversprayed.

With regards to productivity, the data showed that the median was close to the target of 4 h/s/d in the five rounds examined, with marginally higher median productivity observed in 2021 (Fig 4B). This was possible thanks to the close monitoring of field activities using the CIMS for constantly assessing map-sector coverage and daily productivity. The use of real-time monitoring through online dashboards (see S3 Fig) proved highly beneficial in 2021 by allowing field managers to determine where adjustments and corrective actions to boost productivity were needed. The system was also critical for better supervision of spraying performance to identify productivity outliers. This is reflected by the more constant worker output measured in 2021 compared to earlier rounds. In 2017 and 2018, productivity was over dispersed due to some fieldworkers reporting very high daily outputs without proper confirmation of these reports (Fig 4B). The distribution of productivity in 2021 showed a more even spread around the target. Factors affecting worker productivity on Bioko Island are the subject of ongoing research.

A requisite for an effective SDSS is the human resource capacity to enter, use, process and interpret data [22, 23, 35]. During the last two IRS rounds on Bioko Island (2020 and 2021), capacity building of fieldworkers was reinforced and promoted as an essential activity. There was a constant interaction between spray teams and campaign managers through the operational dashboards. Map-sector-level coverage data were used to prompt departure from map-sectors where and when high coverage was achieved. The longer-term goal is to train fieldworkers in identifying issues affecting productivity and coverage and in swiftly responding with corrective actions. Such training is a gradual process that takes time to produce the necessary human capacity to improve IRS deployment. This steep learning curve was another reason why, despite the progress, coverage and productivity indicators in 2021 still showed substantial room for improvement. Guiding a team of over 100 sprayers across thousands of map-sectors to spray tens of thousands of houses in mostly urban areas is a complex task that unavoidably falters. The ultimate goal is a greater operational efficiency, lower underspraying and overspraying, and productivity higher than the minimum threshold of 4 h/s/d (*i.e.* a picture similar to Scenario 4 in Fig 1 and Box 1). Notwithstanding this challenge, the engagement of fieldworkers returned positive outcomes and the trends revealed by the data promise that future rounds will see further advancements.

The fine balance between achieving optimal coverage and using limited resources cost-effectively can be easily altered by the spatial resolution at which coverage is measured. The grid-based coding system at the core of the CIMS [31] (S2 Text and S2 Fig) promotes a highly granular spatial approach to data collection, processing, analysis and feedback. This grid-based approach helps overcome the effects of the MAUP and resource allocation inefficiencies by increasing the spatial granularity of intervention planning and coverage derivation [19]. As is illustrated in Fig 1, the approach is pragmatic, cost-effective and promotes equitable intervention deployment, all made possible through the use of a robust SDSS. Our ability to evaluate coverage on a finer spatial grid revealed the MAUP and that, despite overall coverage falling in 2021 relative to 2017, operational efficiency increased and overspraying decreased when coverage was measured at the map-sector-level.

An important caveat is that the community effect size of interventions could potentially affect planning and deployment. In the case of IRS, there is considerable uncertainty about the coverage levels required to reach community protection. The 80% to 85% optimal recommendation [7] is likely inherited from canons of early efforts of malaria eradication [36] that are supported by limited evidence [13]. Optimal IRS coverage could well be lower, as has been suggested by data from Malawi [14] and similar to the more widely investigated community effects of bed nets [37–41]. More importantly, a spillover community effect is expected around areas with high intervention coverage [42] and this would also influence IRS planning and deployment. For bed nets, community protective effects have been observed within a 300 m radius from intervened areas [43, 44]. However, the evidence of the size and distance of this effect for IRS is also insufficient, and there are lingering questions of whether the map-sectors or a kernel-based spatial average would provide a better understanding of community protection. Map-sectors were not originally established for coverage calculations but rather to fulfill the need to enumerate houses [31]. Hence, they were not motivated by *a priori* considerations about mosquitoes or spillover effects of control. Defining the optimal size of the grid units for deploying vector control is likely context and intervention-specific and is the subject of ongoing operational research on Bioko Island. Data collected through the CIMS are being analyzed using a robust modeling framework to test the effect sizes of different scenarios of interventions, coverage and spillover.

Understanding spatial area effects could help give rise to IRS coverage patterns that would maximize impact. Hypothetically, if spillover effects of IRS were around 200 m at 80% coverage, then planning deployment at high spatial granularity would allow us to guide spraying along a reticular pattern of map-sectors within which 80% of houses were sprayed and which would be separated from the nearest sprayed map-sectors by four unsprayed map-sectors (Fig 5). This would improve protection to all the population, regardless of whether their neighborhood was sprayed or not. Such a setting would save considerable resources as wider areas would be sprayed using the same workforce and amount of insecticide. This is only a simplified example of a plausible scenario. Whatever the spatial configuration required, the high spatial granularity of the CIMS provides the necessary flexibility for the implementation of field activities.

High spatial granularity is also essential for optimally targeting interventions, which is particularly relevant in areas where disease risk is highly heterogeneous [45]. The use of map-sectors for operational planning and implementation guarantees more precise targeting, tracking and monitoring of interventions (see Box 1). In addition, the increased spatial granularity would provide a more flexible framework for temporal intervention targeting and better scheduling of delivery [46]. This flexibility renders the grid-based approach to data collection a powerful asset of the CIMS.

Every malaria control campaign on Bioko Island is now operated and managed through the CIMS, rendering a highly spatially resolved data source. Aside from IRS, some of the campaigns hosted by the CIMS include focal and door-to-door distribution of long-lasting insecticidal nets, focal larval source management, focal malaria screening and treatment, malaria indicator surveys, behavioural change communication, entomological surveillance, insecticide susceptibility monitoring and worker training and supervision. Monitoring and evaluation of all these campaigns are conducted through the CIMS, from budgeting to planning deployment to defining sampling frameworks for surveys, and more. The ability to track individual-level performance has been used for quality assurance and quality control of interventions [6], algorithms that integrate entomological, parasitological and case data are constantly improved to facilitate surveillance and response to malaria outbreaks [47], and outreach training and supportive supervision for malaria case management [48] in public health facilities of Bioko is

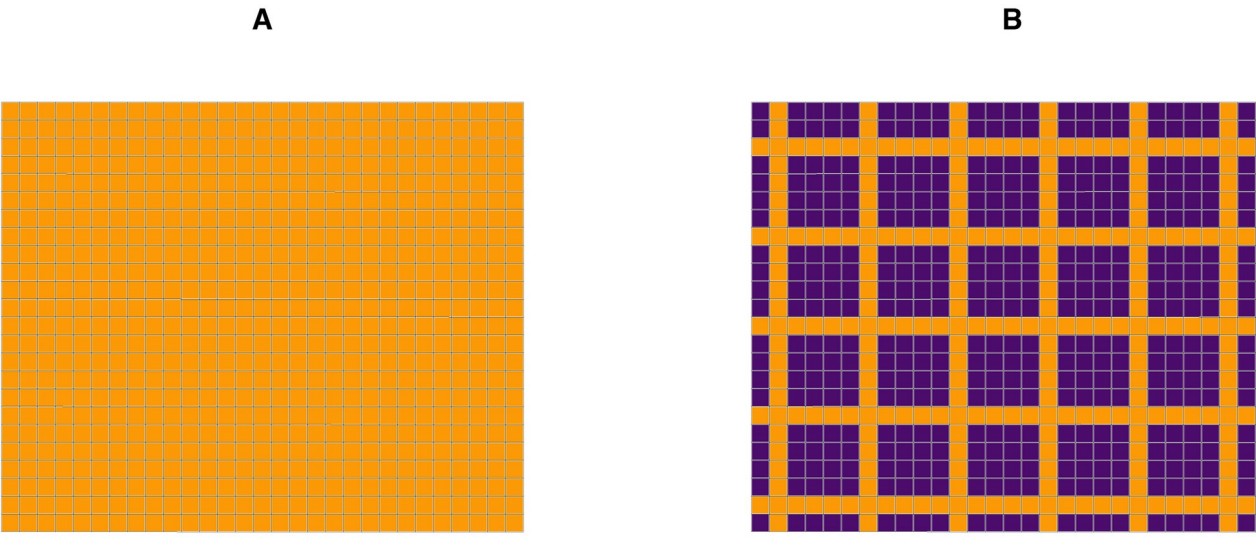

**A** **B**

■ Protected by coverage   ■ Protected by spill over

**Fig 5. Theoretical example of IRS deployment with and without allowing for spillover. A**. All map-sectors within a target population are sprayed at optimal coverage. **B**. Map-sectors are strategically targeted, taking into account a 200 m spillover effect, or the distance comprised by two map-sectors. The spillover effect plausibly wanes with growing distance from high IRS coverage, but for illustrative purposes it is assumed that everyone inhabiting the purple map-sectors is equally protected by the intervention. In **A**, 696 map-sectors are sprayed whereas in **B** only 259 are sprayed.

being established through the CIMS. Importantly, the CIMS offers excellent versatility to adapt to public health interventions beyond malaria vector control and in settings outside of Bioko Island.

## Supporting information

**S1 Text. The need to establish a SDSS to implement IRS operations on Bioko Island.** The file provides more background on the logistics as well as some of the challenges of IRS operations on Bioko Island.
(PDF)

**S2 Text. The CIMS infrastructure.** The file contains technical details of the different components of the CIMS.
(PDF)

**S1 Fig. A simplified schema of the CIMS workflow.**
(PDF)

**S2 Fig. The CIMS grid-based geographical coding system.**
(PDF)

**S3 Fig. Example of IRS coverage dashboard.**
(PDF)

**S4 Fig. Guiding IRS based on refusals.**
(PDF)

## Acknowledgments

We thank the National Malaria Control Program and the Ministry of Health and Social Welfare of Equatorial Guinea, as well as Marathon Oil, Noble Energy, AMPCO (Atlantic Methanol Production Company) and the Ministry of Mines and Energy of Equatorial Guinea for their continued support of malaria control on Bioko Island. We would like to recognize the efforts of the many spray operators who work hard to deliver IRS to protect the people of Bioko Island.

## Author Contributions

**Conceptualization:** Carlos A. Guerra.

**Data curation:** Guillermo A. García, Olivier Tresor Donfack, Carlos A. Guerra.

**Formal analysis:** Carlos A. Guerra.

**Investigation:** Guillermo A. García, Carlos A. Guerra.

**Methodology:** Carlos A. Guerra.

**Project administration:** Guillermo A. García, Wonder P. Phiri, Christopher Schwabe.

**Resources:** Guillermo A. García, Brent Atkinson, Emily R. Hilton, Jeremías Nzamío Mba Eyono, Marcos Mbulito Iyanga, Liberato Motobe Vaz, Restituto Mba Nguema Avue.

**Software:** Brent Atkinson, Emily R. Hilton.

**Supervision:** Guillermo A. García, Wonder P. Phiri, Christopher Schwabe, Carlos A. Guerra.

**Validation:** Guillermo A. García, Jordan M. Smith, John Pollock, Josea Ratsirarson, Edward M. Aldrich, David L. Smith, Christopher Schwabe, Carlos A. Guerra.

**Visualization:** Emily R. Hilton.

**Writing – original draft:** Guillermo A. García, Carlos A. Guerra.

**Writing – review & editing:** Guillermo A. García, Brent Atkinson, Olivier Tresor Donfack, Emily R. Hilton, Jordan M. Smith, Jeremías Nzamío Mba Eyono, John Pollock, Josea Ratsirarson, Edward M. Aldrich, David L. Smith, Christopher Schwabe, Carlos A. Guerra.

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
