## [Decision Letter · Decision Letter 0]

21 Dec 2021

PDIG-D-21-00081

Real-time, spatial decision support to optimize malaria vector control: the case of indoor residual spraying on Bioko Island, Equatorial Guinea

PLOS Digital Health

Dear Dr. Guerra,

Thank you for submitting your manuscript to PLOS Digital Health. After careful consideration, we feel that it has merit but does not fully meet PLOS Digital Health's publication criteria as it currently stands. Therefore, we invite you to submit a revised version of the manuscript that addresses the points raised during the review process.

Three reviewers have provided detailed reviews of your manuscript, raising substantial concerns, as well as providing suggestions for improvements. Please consider carefully all of their comments. Note that Reviewer 2 has also provided comments added onto the PDF of your manuscript (attached). 

We look forward to receiving your revised manuscript.

Kind regards,

Valentina Lichtner

Academic Editor

PLOS Digital Health

Journal Requirements:

1. We ask that a manuscript source file is provided at Revision. Please upload your manuscript file as a .doc, .docx, .rtf or .tex. If you are providing a .tex file, please upload it under the item type ‘LaTeX Source File’ and leave your .pdf version as the item type ‘Manuscript’.

2. Please provide us with a direct link to the base layer of the map used in Figure 1 and ensure this location is also included in the figure legend. 

Please note that, because all PLOS articles are published under a CC BY license (creativecommons.org/licenses/by/4.0/), we cannot publish proprietary maps such as Google Maps, Mapquest or other copyrighted maps. If your map was obtained from a copyrighted source please amend the figure so that the base map used is from an openly available source.

Please note that only the following CC BY licences are compatible with PLOS licence: CC BY 4.0, CC BY 2.0 and CC BY 3.0, meanwhile such licences as CC BY-ND 3.0 and others are not compatible due to additional restrictions. If you are unsure whether you can use a map or not, please do reach out and we will be able to help you. 

The following websites are good examples of where you can source open access or public domain maps:

Additional Editor Comments (if provided):

Reviewers' comments:

Reviewer's Responses to Questions

**Comments to the Author**

1. Does this manuscript meet PLOS Digital Health’s publication criteria? Is the manuscript technically sound, and do the data support the conclusions? The manuscript must describe methodologically and ethically rigorous research with conclusions that are appropriately drawn based on the data presented.

Reviewer #1: Yes

Reviewer #2: Yes

Reviewer #3: Partly

2. Has the statistical analysis been performed appropriately and rigorously?

Reviewer #1: Yes

Reviewer #2: Yes

Reviewer #3: I don't know

3. Have the authors made all data underlying the findings in their manuscript fully available (please refer to the Data Availability Statement at the start of the manuscript PDF file)?

Reviewer #1: Yes

Reviewer #2: Yes

Reviewer #3: Yes

4. Is the manuscript presented in an intelligible fashion and written in standard English?

Reviewer #1: Yes

Reviewer #2: Yes

Reviewer #3: Yes

5. Review Comments to the Author

Reviewer #1: This paper is technically satisfactory as far as it goes but in my opinion falls far short of what might be expected for a paper dealing with the development of a spatial decision support system (SDSS). It might be fairer to say that this paper focuses simply on the development of a GIS to manage a spraying programme.

1. I would like to see more discussion of the benefits derived from different spraying scenarios at the map sector level given the apparent spatial spillover effects associated with spraying (as remarked by the authors). The system might be used to test these spatial spillover benefits and how different geographical patterns of spraying might be used to provide the most cost effective treatment. This leads to my second and related point:

2. This paper confuses outputs and outcomes. The purpose of a spraying programme is to reduce malaria (outcome) not simply to manage a spraying programme (output). For this reason the system needs to include malaria data at the finest available space-time scale so that some attempt can be made, within the SDSS, to evaluate the effectiveness of any programme in relation to the outcome which is the aspect of this that matters. This of course presents a significant evaluative challenge but that is what an SDSS is meant to address - otherwise it might just as well be called a GIS which is constructed for the sole purpose of managing and displaying spatial and spatial-temporal data about where and when spraying was carried out.

As a final point more attention needs to be paid to the maps in Figure 1. The choice of colour palette makes it very difficult for the reader to see where the different types of households are located.

Reviewer #2: Dear author:

I read your manuscript entitled: “Real-time, spatial decision support to optimize malaria vector control: the case of indoor residual spraying on Bioko Island, Equatorial Guinea”, and I understand that you using a spatial decision support system (SDSS) to handle the residual spraying on Bioko Island and to optimize malaria vector; and I will try to discuss the paper from a health geography point view, first of all, any geographic research is either:

• Thematic research focuses on how a natural, human or environmental phenomenon appears and spreads, and how is it linked to some geographical location and factors, as it depends mainly on the use of spatial, statistical, and geostatistical analysis tools.

• Territory management: for this approach the researcher used a specific technology to solve a territory problem (like malaria in your case), however, in this approach the problem is not important and researchers don’t focus on the situation and the problem (disease, flood, fires), but the aim is how to use the selected technique to solve this problem in a specific location.

• Methodological research is another level where the researchers try to produce new methods, tools, approaches or optimize the existing ones.

For your paper I suppose it’s A territory management paper, because the paper doesn’t discuss malaria or the causes of contamination and the spatial analysis of malaria, but it discusses “How to handle the malaria situation” so I will go with this approach and here are the suggested notes to make your article better (by the way, it’s a good idea and you make a great academic contribution), most of the notes are for formatting.

• Make your abstract more in-depth and more technical.

• You can insert a literature review about using SDSS in health, and some background about malaria disease and the spray operation.

• You have to make the problem more explicit.

• Explain technics and methods used in the paper in depth especially the architecture of the spray operation, software, tools, server, phone and user side…etc.

• Your result is more like a simple text and just describes and rewrites the results and table without any further technical and statistical details.

• Please rewrite your article using IMRAD structure (introduction, method, result, and discussions), I see that you put the result just after the introduction and you push the data and methods section to the end of the article, I think this is not suitable for an academic article.

Finally, please check the attached PDF file, I include some notes and comments. Good luck!

Reviewer #3: The authors present an interesting application of a SDSS to support an IRS implementation.

I was left a little confused by the aims of the paper: is it to “Describe the use of a novel SDSS to support malaria vector control?” (lines 49 to 50)? Or does the paper present changes to IRS overtime as reported within a bespoke SDSS? If it is the former, I was underwhelmed as to how the SDSS was actually used to support the IRS operations. I think more detail and a clearer picture on how the system is actually used would be of interest to the reader, rather than simply a presentation of the changes in “over-sprayed” and “under-sprayed” map-sectors. 

Some specific areas where more detail on how the SDSS is actually used include:

The abstract reads “High spatial granularity during planning and deployment together with closer follow-up of field teams using real-time data supported more homogeneous delivery of optimal coverage while sustaining high productivity.”, however in the main text of the manuscript I have failed to read or understand how the SDSS was practically used to support an IRS operation. Further clarification (in perhaps the Materials and Methods and the Discussion) would be of interest. 

The Discussion refers to real-time monitoring through online dashboards (line 158). How were these dashboards used in an SDSS to support the IRS operation? Was this a standardized process incorporated into operation based on the SDSS? Can the reader see a snapshot of what the real-time dashboard monitoring interface looks like? Some detail in the Materials and Methods would be informative in regards to monitoring IRS operations using the SDSS. 

I was left confused about a number of aspects of the SDSS and found the materials and methods section lacking sufficient detail to understand the operational processes involved in using the SDSS to support IRS:

• How are baseline household data recorded and updated into the SDSS? 

• How are IRS data updated into the SDSS? The paper states it is real-time monitoring? How are these data reported to the SDSS (via mobile device? By the Spray Teams?) 

• What additional attribute data are recorded into the SDSS? 

 o Is there any information collected or reported as to why households are not sprayed?

• Can the grid-based coding system be described in a little more detail?

• I believe some of this is described in the supplementary material and may be referenced, however for context I think some of this information should be included in the main text also.

Does the SDSS have capacity to facilitate follow-up spraying operations in under-sprayed areas as a means to improve coverage and achieve optimal outcomes?

Also – who is the user of the SDSS? Field Managers? Overall Supervisors? Other people? Is there a high level of acceptability of the SDSS among the intended users? 

It is an interesting argument in the Discussion that “operational efficiency increased significantly between 2017 to 2021 and that this improvement was attained by “significantly reducing overspraying” (Lines 127 to 129). This seems like somewhat of a controversial and perhaps questionable operational decision – particularly when discussing “equitable” delivery of interventions?? 

• How does a spray-person decide whose house not to spray within a map-sector? Surely this could create operational issues within a community? If a household requests spraying – are they given this intervention? 

• Is there evidence to suggest that over-spraying in certain “map-sectors” contributes to under-spraying elsewhere? And why so? Is it just simply a limited lack of resources and they run out of “insecticide”, or could there be other underlying factors inhibiting spray coverage in certain areas?

• Are there spatial patterns to underspraying? i.e. areas consistently undersprayed? Were data collected in regards to why certain areas were undersprayed? Can this information be accessed via the SDSS?

6. PLOS authors have the option to publish the peer review history of their article (what does this mean?). If published, this will include your full peer review and any attached files.

**Do you want your identity to be public for this peer review?** For information about this choice, including consent withdrawal, please see our Privacy Policy.

Reviewer #1: Yes: Emeritus Professor Robert Haining

Reviewer #2: Yes: Belkacem Lahmar

Reviewer #3: No

---

## [Editor Report · Decision Letter 1]

4 Mar 2022

PDIG-D-21-00081R1

Real-time, spatial decision support to optimize malaria vector control: the case of indoor residual spraying on Bioko Island, Equatorial Guinea

PLOS Digital Health

Dear Dr. Guerra,

Thank you for revising your manuscript and answering reviewers' comments. We find the revision address the reviewers' concerns. However as editors of this Journal, we ask you to please restructure the manuscript placing the Methods before the Results, as traditionally found in most research papers and as also suggested by one of the reviewers. 

We look forward to receiving your revised manuscript.

Kind regards,

Valentina Lichtner

Academic Editor

PLOS Digital Health
---

## [Editor Report · Decision Letter 2]

15 Mar 2022

Real-time, spatial decision support to optimize malaria vector control: the case of indoor residual spraying on Bioko Island, Equatorial Guinea

PDIG-D-21-00081R2

Dear Dr Guerra,

We are pleased to inform you that your manuscript 'Real-time, spatial decision support to optimize malaria vector control: the case of indoor residual spraying on Bioko Island, Equatorial Guinea' has been provisionally accepted for publication in PLOS Digital Health.

Thank you for restructing the paper as requested - we are aware the guidelines on our site give authors the option to structure their manuscripts in different ways.

Best regards,

Valentina Lichtner

Academic Editor

PLOS Digital Health